

# Identifying risk zones and landscape features that affect common leopard depredation

Thakur Silwal[1,2], Bijaya Neupane[2,3], Nirjala Raut[2], Bijaya Dhami[4], Binaya Adhikari[5], Amit Adhikari[2], Aakash Paudel[2], Shalik Ram Kandel[2] and Mahamad Sayab Miya[6]

[1] Tribhuvan University, Institute of Forestry, Kathmandu, Nepal
[2] Tribhuvan University, Institute of Forestry, Pokhara, Nepal
[3] Department of Forest Sciences, Faculty of Agriculture and Forestry, University of Helsinki, Helsinki, Finland
[4] Department of Biological Sciences, University of Alberta, Edmonton, Canada
[5] Department of Biology, University of Kentucky, Lexington, KY, United States of America
[6] Department of Biology, Western Kentucky University, Bowling Green, KY, United States of America

Corresponding author
Thakur Silwal, tsilwal@iofpc.edu.np

## ABSTRACT

Human-wildlife conflict (HWC) is a pressing issue worldwide but varies by species over time and place. One of the most prevalent forms of HWC in the mid-hills of Nepal is human-common-leopard conflict (HLC). Leopard attacks, especially in forested areas, can severely impact villagers and their livestock. Information on HLC in the Gorkha district was scarce, thus making it an ideal location to identify high-risk zones and landscape variables associated with such events. Registered cases were collected and reviewed from the Division Forest Office (DFO) during 2019-2021. Claims from DFO records were confirmed with herders and villagers via eight focus group discussions. To enhance modeling success, researchers identified a total of 163 leopard attack locations on livestock, ensuring a minimum distance of at least 100 meters between locations. Using maximum entropy (MaxEnt) and considering 13 environmental variables, we mapped common leopard attack risk zones. True Skill Statistics (TSS) and area under receiver-operator curve (AUC) were used to evaluate and validate the Output. Furthermore, 10 replications, 1,000 maximum iterations, and 1000 background points were employed during modeling. The average AUC value for the model, which was $0.726 \pm 0.021$, revealed good accuracy. The model performed well, as indicated by a TSS value of $0.61 \pm 0.03$. Of the total research area ($27.92 \text{ km}^2$), about 74% was designated as a low-risk area, 19% as a medium-risk area, and 7% as a high-risk area. Of the 13 environmental variables, distance to water (25.2%) was the most significant predictor of risk, followed by distance to road (16.2%) and elevation (10.7%). According to response curves, the risk of common leopard is highest in the areas between 1.5 to 2 km distances from the water sources, followed by the closest distance from a road and an elevation of 700 to 800 m. Results suggest that managers and local governments should employ intervention strategies immediately to safeguard rural livelihoods in high-risk areas. Improvements include better design of livestock corrals, insurance, and total compensation of livestock losses. Settlements near roads and water sources should improve the design and construction of pens and cages to prevent livestock loss. More studies on the characteristics of victims are suggested to

enhance understanding of common leopard attacks, in addition to landscape variables. Such information can be helpful in formulating the best management practices.

## INTRODUCTION

Human-wildlife conflict (HWC) results from negative interactions between humans and wildlife in shared landscapes (*Silwal, Kolejka & Sharma, 2016*; *Silwal et al., 2017*; *Mukeka et al., 2019*). Large carnivores tend to amplify the situation due to their extensive territorial needs and dietary requirements (*Dickman, 2009*; *Redpath et al., 2013*; *Chetri et al., 2019*), presenting significant challenges for villagers and their livestock (*Khorozyan et al., 2015*; *Boronyak, Jacobs & Wallach, 2020*). Coexistence in human-altered landscapes often leads to adverse outcomes such as livestock depredation, human casualties, and retaliatory killing of carnivores, thus intensifying conflicts among stakeholders and reducing overall support for conservation (*Inskip & Zimmermann, 2009*; *Pooley et al., 2017*). Human-induced conflicts threaten the long-term survival of carnivores, substantially contributing to their population decline (*Adhikari et al., 2022a*). The escalation of HWC is a global concern, although the species involved in these incidents vary across different regions. In Africa, conflicts primarily include lions, leopards, hyenas, cheetahs, and wild dogs in human-large carnivore conflict scenarios (*Koziarski, Kissui & Kiffner, 2016*; *Bhandari, Baral & Adhikari, 2022*). Conversely, in Europe, the grey wolf, Eurasian lynx, and brown bear are responsible for HWC (*Cimatti et al., 2021*). Similarly, in Asia, conflicts predominantly arise from tigers, common leopards, snow leopards, and Himalayan black bears (*Sangay & Vernes, 2008*). In Nepal, the common leopard (*Panthera pardus*), locally known as "*chituwa*," is the source of many conflict incidents, especially in the mid-hills region (*Adhikari et al., 2020*; *Kandel et al., 2023*).

The common leopard, classified as a vulnerable habitat-generalist species by the IUCN Red List in Nepal, thrives across diverse habitats, ranging from lowlands to elevations of 4400 m, encompassing grasslands, forests, and mountainous terrains (*Jnawali et al., 2011*; *Can et al., 2020*; *Baral et al., 2023*). Typically, common leopards occupy home ranges spanning 8 to 15 km$^2$, with larger territories in arid and semi-arid regions due to limited prey availability (*Bothma et al., 1997*; *Odden et al., 2014*). Their diet consists of many prey species, including birds, rodents, and other small mammals, yet they favor medium-sized ungulates as their primary food source (*Hayward et al., 2006*; *Bhandari, Baral & Adhikari, 2022*).

Human common-leopard conflict (HLC) is a critical issue across leopard-range Africa (*Constant, Bell & Hill, 2015*; *Viollaz, Thompson & Petrossian, 2021*) and South Asia, including India (*Athreya et al., 2015*), Pakistan (*Kabir et al., 2014*), and Nepal (*Baral et al., 2022b.*; *Adhikari et al., 2022b*; *Kandel et al., 2023*). The success of conservation programs such as community forests in mid-hills has increased the common leopard population,
resulting in a surge of HLC incidents (*Ghimirey, 2006; Dhungana et al., 2016; Baral et al., 2021b*). These conflicts impact rural livelihoods significantly since common leopards are major predators of livestock, responsible for about 78% of all losses in Nepal in addition to attacks on grazing livestock such as goats and sheep near protected areas (*DNPWC, 2017; Lamichhane et al., 2018; Dhungana et al., 2022; Shahi et al., 2022*). For example, villages adjacent to Chitwan National Park (CNP) and Annapurna Conservation Area in Nepal often suffer from leopard depredation on dogs, goats, and cattle, greatly impacting the livelihoods of local people (*Baral et al., 2022a*).

Species distribution models such as maximum entropy (MaxEnt) are often used to predict wildlife habitat distribution (*Thorn et al., 2009; Clements et al., 2012; Adhikari et al., 2023; Dhami et al., 2023a; Dhami et al., 2023b*) and estimate species density across the landscape (*Phillips, Anderson & Schapire, 2006*). It has been used to model species distribution and ecological niches since it requires presence-only data and environmental predictors, reducing bias and elevating accuracy (*Phillips, Anderson & Schapire, 2006*). Studies by *Elith et al. (2011)* and *Wisz et al. (2008)* indicated that this modeling software exhibited the highest level of predictive accuracy compared to other modeling software, even when the sample sizes were relatively small.

Several scholars have used MaxEnt for HWC risk zone mapping (*Sharma et al., 2020a; Sharma et al., 2020b; Adhikari et al., 2022a; Khosravi et al., 2023*). Most HWC research has focused on quantifying damage, wildlife species involved, and human perceptions of conflicts (*Treves et al., 2006; Silwal, Kolejka & Sharma, 2016; Silwal et al., 2017; Ruda, Kolejka & Silwal, 2018; Ruda, Kolejka & Silwal, 2020; Shahi et al., 2022; Silwal et al., 2022*). This study explored a new dimension of MaxEnt by analyzing common leopard depredation events in the Gorkha district for three years (2019-2021) to create a three-way classification (high, medium, and low) of conflict risk zones. In addition, an attempt was made to determine the major factors (topographic, vegetation, and anthropogenic) that might be responsible for livestock depredation by common leopards.

## MATERIALS AND METHODS

### Study area

This study was conducted in Gorkha Municipality and Bhimsen Thapa Rural Municipality of Gorkha district (*i.e.,* the epicenter of the devastating 2015 earthquake), located in the mid-hills of Nepal (Fig. 1). Gorkha was selected because *Adhikari et al. (2022a)* identified this location as a high-risk HWC zone. The district is located in Gandaki Province with latitudes 27°15′N to 28°45′N and longitudes 84°27′E to 84°57′E. The total area is 3614.70 km$^2$, while that of Gorkha Municipality (*i.e.,* district headquarters) is 131.56 km$^2$ and Bhimsen Thapa Rural Municipality is 101 km$^2$ with forest coverage of 33% and 39.1%, respectively (*DFRS, 2018*). The average altitudinal range of the district is 228 m (*e.g.,* banks of Marsyangdi River) above sea level to 8163 m (*e.g.,* Mt. Manaslu Himalaya). Annual rainfall in the district ranges from 0 mm in December to 529.4 mm in July. Similarly, the maximum temperature (31.9 °C) is observed during June and the minimum (6 °C) in January.

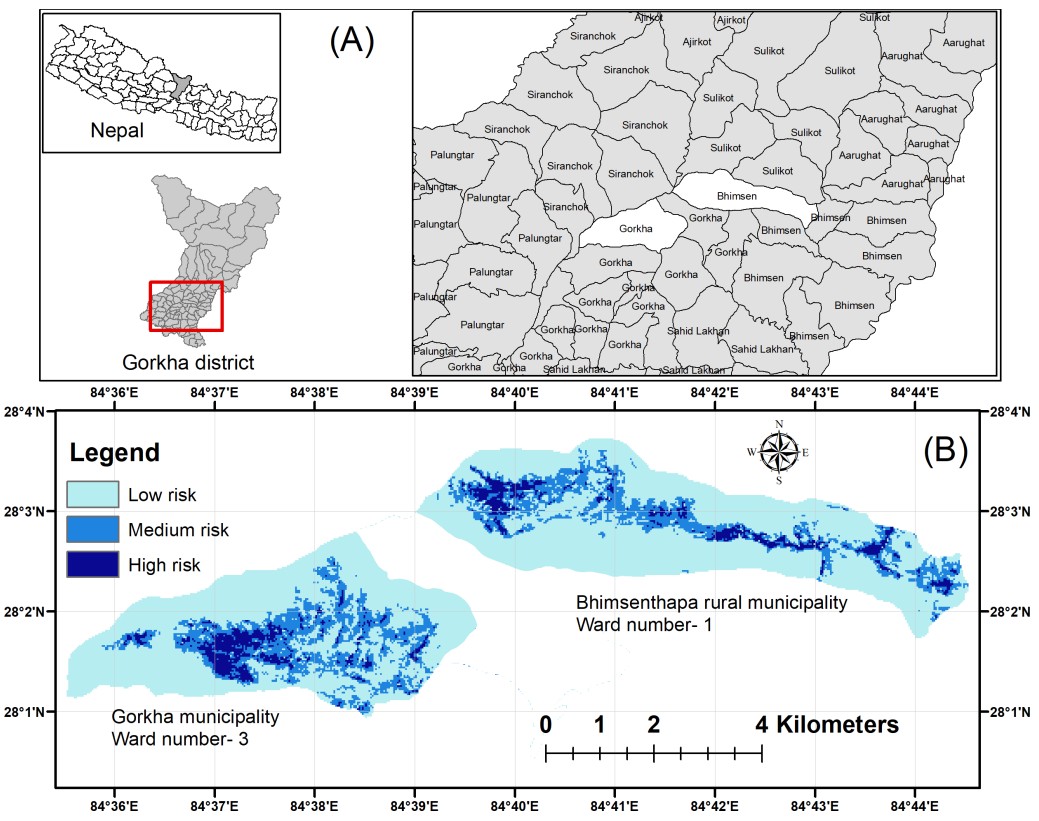

**Figure 1** Map showing the study area (A) Location of Gorkha district and study sites (*FRTC, 2019*), and (B) Categories of common leopard attack risk zones. Image source credit: Open data Nepal.

According to *Central Bureau of Statistics (2022)*, the total population of Gorkha Municipality is 53,285 people (15,298 households), and Bhimsen Thapa Rural Municipality is 17,118 people (5,265 households). Major ethnic groups in the study area include *Brahmin, Chettri, Newar, Darai, Magar, Damai, Kami, and Kumal*, which mainly belong to the Hindu religion. Their primary livelihood strategy is agriculture, especially farming and animal husbandry. Common livestock include buffalo (*Bubalus bubalis*), domestic goats (*Capra aegagrus hircus*), and poultry. The relative abundance of plants in the district are *Shorea robusta* in the southern aspect and *Schima- Castanopsis* forest in the northern aspect, with other associated species such as *Syzygium cumuni*, *Rhus* spp., *Terminalia alata, and Semicarpus anacardium* in the lower altitudes, whereas *Myrica esculanta*, *Pinus* spp. and *Madhuca indica* in the higher altitudes. Predominant wildlife species include the common leopard, jungle cat (*Felis chaus*), golden jackal (*Canis aureus*), yellow-throated marten (*Martes flavigula*), Rhesus macaque (*Macaca mulatta*), barking deer (*Cervus vagianalis*), and Indian crested porcupines (*Hystrix indica*) and so forth.

## Research design

Prior to conducting any fieldwork, the Government of Nepal, Ministry of Forests and Environment, Department of Forests and Soil Conservation, in co-ordination with

Divisional Forest Office (DFO), Gorkha issued research permissions for the study (issued # 1728-2078/2079). The government has assigned responsibility to verify cases for providing relief to wildlife victims and dealing with issues outside protected areas. From 2019-2021, injuries and fatal cases of common leopard attacks on people and livestock in most affected areas were considered after consultation with the DFO. Meetings of concerned people were organized, including focus group discussions, and recorded locations and topographic, vegetation, environmental, and anthropogenic variables were used as primary data for each attack site. Secondary sources of information included DFO documents, the Manaslu Conservation Area, and other literature.

## Data collection
### Focus group discussion
Initially, focus group discussions (FGD) were conducted with the DFO staff, Manaslu Conservation Area staff, and the Federation of Community Forestry Users Nepal to identify the most affected HLC sites in Gorkha. During the FGD, participants gave district-wide field-based updates on the intensities of HLC. The FGDs were followed by a review of registered cases of livestock depredation by common leopards during 2019-2021 since victims had to file claims for government relief through the DFO Office in Gorkha. Then, wards 3 and 1 of Gorkha Municipality and Bhimsen Thapa Rural Municipality were selected for the detailed field survey. During March 2022, all possible risk zones were visited based on closeness to the livestock shed and grazing areas. Herders were absent when livestock depredation occurred. After a list of herders was compiled at the sites, we conducted eight FGDs (four at each location) consisting of five to eight herders and three to five villagers. Participants at some workshops were replicated to ensure that the information on livestock depredation by common leopards would not be missed or repeated while mapping the severity of attacks across sites.

### Recording locations of livestock depredation
After the FGDs, 163 incidences of livestock depredation were recorded at both sites (grazing land and sheds) during the study period using a Geographic Position System (GPS) device. Each GPS location was considered unique since they were spaced at least 100 m between the two study sites (*Silwal et al., 2017*; *Ruda, Kolejka & Silwal, 2020*; *Karki & Panthi, 2021*). Since our environmental, anthropogenic, and topographic variables had a resolution of 100 m*100 m, no two events had corresponding values, thereby minimizing bias due to spatial autocorrelation.

### Topographic variables
Common leopards' spatial distribution is primarily governed by topography (*Kandel et al., 2023*). These variables were used previously to model the habitat and risk zones of common leopards and other carnivores in Nepal (*Bista, Panthi & Weiskopf, 2018*; *Adhikari et al., 2021*). A Digital Elevation Model (DEM) with a 30 m resolution was downloaded from the United States Geological Survey database (*USGS, 2022*), while the slope and aspect were derived using ArcGIS software (*ESRI, 2020*). The land use land cover map of the study area was collected from the Forest Research and Training Center, Government

**Table 1  Environmental variables used for modeling.**

| Source | Category | Variable | Unit |
|---|---|---|---|
| USGS | Topographic | Elevation | M |
|  |  | Aspect | Degree |
|  |  | Slope | Degree |
| GEOFABRIK |  | Distance to water source | M |
| MODIS | Vegetation-related | Mean EVI | Dimensionless |
|  |  | Maximum EVI | Dimensionless |
|  |  | Minimum EVI | Dimensionless |
|  |  | Standard deviation of EVI | Dimensionless |
| GFC |  | Forest | Dimensionless |
| GEOFABRIK | Anthropogenic | Distance to motor road | M |
|  |  | Distance to path | M |
|  |  | Distance to settlements | M |
| ICIMOD |  | Land use/land cover | M |

of Nepal (*FRTC, 2019*). Shapefiles of water sources were downloaded from the Geofabrik website (*GEOFABRIK, 2022*) and converted to distance raster files using ArcGIS (*ESRI, 2020*).

## Vegetation related variables

The diet of common leopards consists primarily of wild and domestic herbivores (*Devkota, Silwal & Kolejka, 2013*; *Devkota et al., 2017*), underscoring the need for vegetation (*Andersen et al., 2000*). Therefore, the forest cover of Global Forest Change (GFC) was downloaded from Earth engine partner Appspot and was used as a vegetable variable (*Hansen et al., 2013*). In addition, the Enhanced Vegetation Index (EVI) time series image was downloaded from the Moderate Resolution Imaging Spectroradiometer (MODIS) sensor from the USGS–EVI was used to model possible common leopard attacks. Afterward, ENVI software was used to smooth the data using an adaptive Savitzky-Golay filter in TIMESAT (*Jönsson & Eklundh, 2004*), which decreased the cloud effect, thus allowing us to obtain EVI mean, maximum, minimum, and standard deviation.

### Environmental variables used in modeling

To utilize maximum entropy (MaxEnt) modeling, data on topography, vegetation, and anthropogenic variables (Table 1) was gathered from various sources. The environmental variables in the models have been used for habitat suitability mapping (*Sharma et al., 2020a*) and similar risk zone mapping studies (*Karki & Panthi, 2021*).

### Anthropogenic variables

Large numbers of livestock depredations were recorded at both study sites due to the proximity of human settlements that overlap the native range of common leopards (*Chetri et al., 2019*; *Sijapati et al., 2021*). Thus, assessing anthropogenic causes that led to common leopard depredation on livestock was crucial since these variables can be controlled. Project success was defined in terms of common leopard conservation and reducing

livestock mortality due to depredation. Distance to paths (trails used by livestock and humans), roadways (vehicle roads), and settlements were used as anthropogenic variables. Geofabrik's website was used to extract data about human paths, roads, and buildings (*GEOFABRIK, 2022*). The Nepalese Department of Survey provided data on settlements, and a distance raster file was created using ArcGIS10.8.1 (*ESRI, 2020*). The national land use and cover database of Nepal used public domain Landsat TM data of 2010 and replicable methodology (*Uddin et al., 2015*).

## Data analysis

The MaxEnt software was used to generate a predicted livestock depredation risk map using geo-referenced incident points of common leopard attacks and environmental variables as input variables (Table 1) (*Elith et al., 2006*; *Phillips, Anderson & Schapire, 2006*; *Phillips et al., 2017*). The area under the receiver-operator curve (AUC) (*Pearce & Ferrier, 2000*) was used to validate the model, and true skill statistics (TSS) was used to evaluate it (*Allouche, Tsoar & Kadmon, 2006*). The AUC value of 1.0 indicates the perfect model, while 0.9−0.99, 0.8−0.89, 0.7−0.79, and 0.51−0.69 indicate excellent, good, fair, and poor model fit (*Hanley & McNeil, 1982*; *Carter et al., 2016*). The TSS value ranges between −1 and 1, where the higher value shows the model's good performance (*Allouche, Tsoar & Kadmon, 2006*). Multicollinearity was found to be less than 0.7 between the variables, which was acceptable for modeling (*Dormann et al., 2013*). Seventy percent of the data was utilized to train the model, while the remaining thirty percent was used to validate it. We employed 10 replications, 1,000 maximum iterations, and 1,000 background points during modeling based on *Barbet-Massin et al. (2012)*. Thresholds to optimize the sum of specificity and sensitivity were utilized (*Liu, Newell & White, 2016*) to compute TSS and construct the binary map from the continuous one.

## RESULTS

### Distribution of risk area

Of the study area (27.92 km$^2$), 20.67 km$^2$ (74%) was identified as a low-risk zone, 5.21 km$^2$(19%) as a medium-risk zone and 2.04 km$^2$(7%) as a high-risk zone. Within the high-risk zone, Gorkha Municipality ward number 3 constituted 1.1 km$^2$, whereas Bhimsen Thapa Rural Municipality ward number 1 constituted 0.94 km$^2$ (Fig. 1B).

### Important environmental variables predicting the common leopard risk zone

The most important variable for predicting common leopard risk zones was the distance to water (25.2%), followed by distance to road (16.2%) and elevation (10.7%). Similarly, the distance to settlement (17.6%) and distance to the road (14.7%) have high permutations, reinforcing their significance in the model (Table 2, Fig. 2).

Response curves (Fig. 3) indicated that the risk of common leopard conflict peaked at the nearest distance from the road and human/livestock path, and vice-versa. The curves also indicated that the HLC risk was highest between 1.5 and 2 kilometers from water sources and between 700 and 800 m in elevation.

**Table 2  Percentage contribution and permutation importance of each variable to develop the model.**

| Variable | Percent contribution | Permutation importance |
|---|---|---|
| dist_water | 25.2 | 8.5 |
| dist_road | 16.2 | 17.6 |
| elevation | 10.7 | 13.3 |
| dist_path | 8.2 | 10 |
| lulc | 8.2 | 4 |
| dist_settle | 7.7 | 14.7 |
| aspect | 6.1 | 9 |
| evi_sd | 4.7 | 5.6 |
| slope | 3.7 | 5.5 |
| evi_min | 3.6 | 4.4 |
| forest | 2.6 | 2.9 |
| evi_max | 1.6 | 2.9 |
| evi_mean | 1.5 | 1.8 |

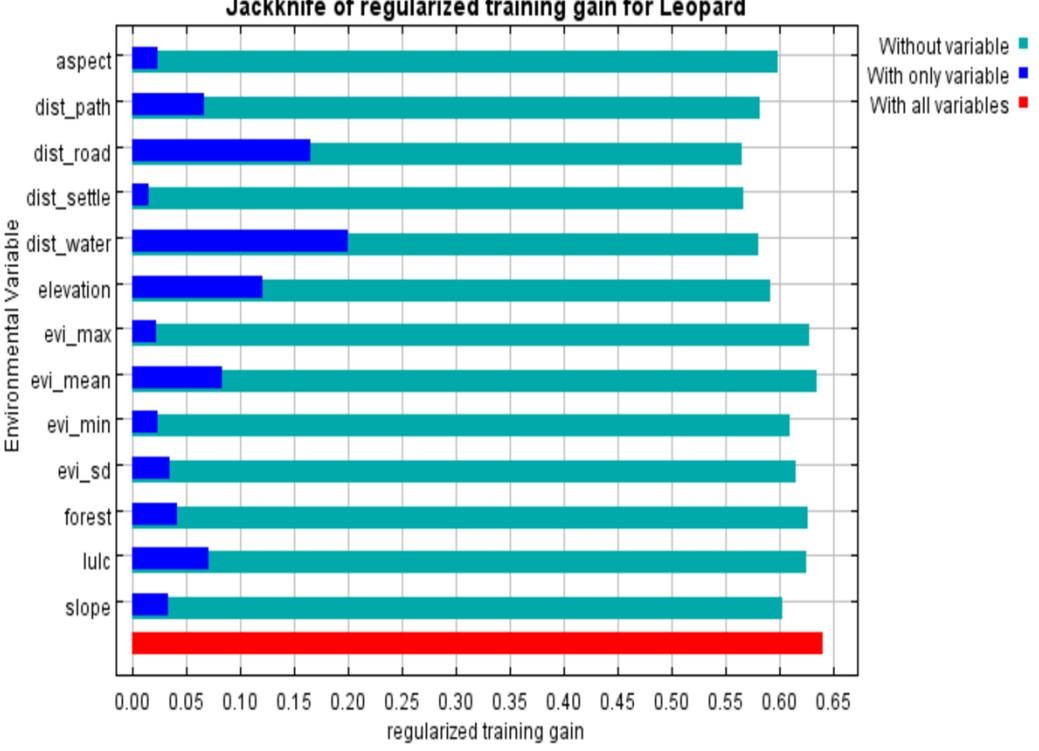

**Figure 2  Jackknife regularized training gain for measuring variable importance.**

## Model accuracy

The model's accuracy was fair, with an average AUC value of $0.726 \pm 0.021$. The TSS value was $0.61 \pm 0.03$, which indicates that the model has some discriminatory power because the TSS value ranges between $-1$ and $1$.
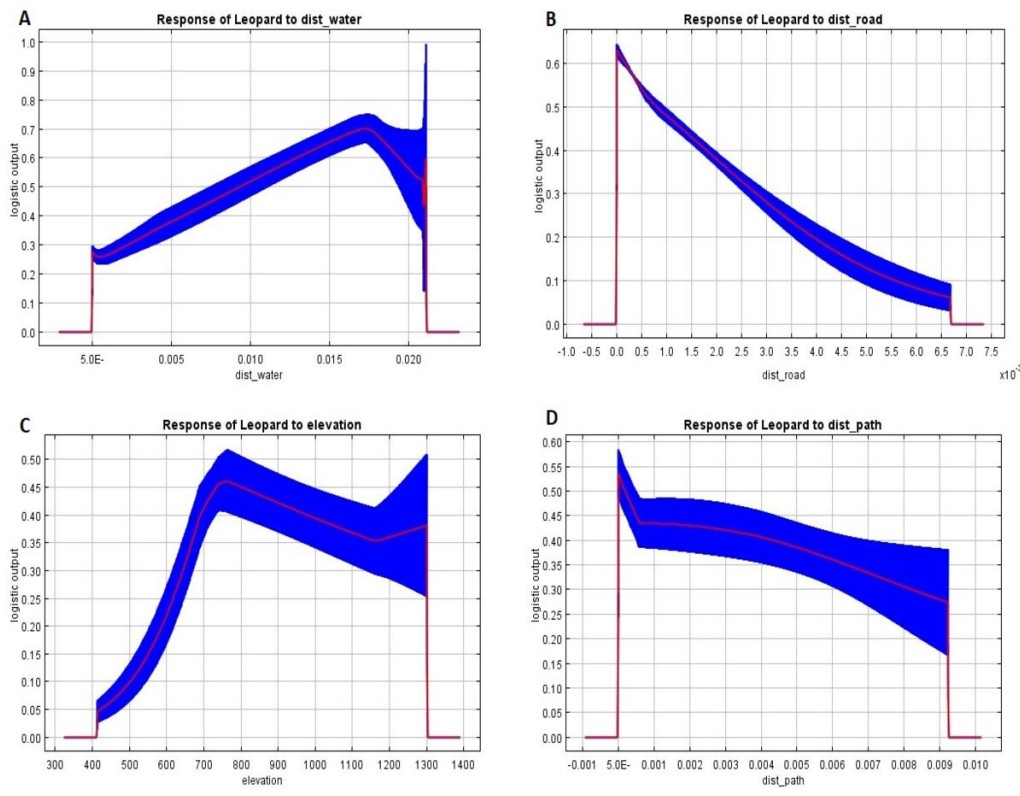

**Figure 3** Response curves indicating the responses of variables towards predicting risk zones.

## DISCUSSION

This study predicted HLC risk zones and shed light on important landscape features that influence attacks on livestock in the Gorkha district, mid-hills of Nepal. It will aid in developing suitable mitigation strategies by informing the Division Forest Office and local governments.

### Risk zones with respect to common leopard attacks

Our study identified a 2.04 km$^2$ area as a high-risk zone for HLC in Gorkha district, with Gorkha Municipality ward number 3 constituting a significant 1.1 km$^2$ of this zone. This finding aligns with a parallel study in the neighboring Aadhikhola Rural Municipality (Syangja district), which pinpointed a 2.76 km$^2$ very high-risk zone and a 5.56 km$^2$ high-risk zone for common leopard attacks (*Adhikari et al., 2021*). Moreover, *Adhikari et al. (2022a)* categorized the Gorkha district as a high-risk zone for human-induced common leopard mortality. This classification underscores the pressing need for targeted conservation efforts and management strategies within the region. Identifying high-risk zones provides valuable insights for devising intervention strategies for mitigating HLC while safeguarding villagers and common leopard populations in Gorkha and its neighboring areas.

Potentially, a contributing factor to this observed pattern could be linked to the comparatively low vegetation cover in Gorkha Municipality (33.0%), resulting in lower

prey density within the forest in contrast to Bhimsen Thapa Rural Municipality (with 39.1% forest cover) (*DFRS, 2018*). Recent research has highlighted a concerning decline in several prey species vital to common leopards, such as barking deer and wild boar, particularly in the mid-hill regions (*Baral et al., 2021a*; *Baral et al., 2021b*). This decline may force common leopards to seek alternative food sources, as *May (1977)* suggested, including preying on livestock and even dogs (*Athreya et al., 2016*). An increase in common leopard attacks near settlements and agricultural lands in the Gandaki province of Nepal was reported by *Baral et al. (2021b)*, emphasizing the growing interaction between common leopards and human-inhabited areas. Additionally, *Rostro-García et al. (2016)* noted that the prevalence of human settlements, mainly those dependent on animal husbandry, may have increased livestock density, resulting in heightened livestock kills by common leopards in Bhutan. As an outcome, areas inhabited by humans are emerging as crucial zones for wildlife conflict in Nepal (*Acharya et al., 2016*). Moreover, events such as livestock depredation and severe injuries to humans generate negative attitudes toward marauding wildlife (*Treves & Karanth, 2003*; *Bagchi & Mishra, 2006*). Such negative sentiments can lead to the killing or at least favor the killing of wildlife species (*Don Carlos et al., 2009*), thereby undermining efforts to conserve apex predators in the mid-hills (*Baral et al., 2022b.*). This interconnected chain of factors underscores the complexity of human-common leopard interactions in these regions and emphasizes the importance of integrated conservation strategies.

## Important environmental variables predicting the common leopard risk zone

Key predictors of common leopard risk zones in our study included distance to water, road distance, and elevation. The study by *Sharma et al. (2020a)* identified crucial factors for HWC in the Kanchenjunga landscape, with distance to road, elevation, livestock density, and mean annual temperature as good predictors. In contrast, *Upadhyaya et al. (2020)* emphasized the importance of livestock numbers, ethnic groups, and village distance from the park boundary for explaining livestock losses in Bardia National Park, Nepal. Likewise, *Sijapati et al. (2021)* found that common leopard-induced livestock kills in Bardia National Park were more prevalent at locations distant from the park boundary, subject to seasonality. Key factors of common leopard depredation in South Africa included distance to villages, followed by distance to water, roads, nature reserve, and elevation (*Constant, Bell & Hill, 2015*).

The risk of common leopard conflict in Gorkha was highest between 1.5–2 Km from water sources. Water attracts domestic and wild animals, thus increasing the number of leopard attacks due to advantageous hunting conditions near these areas. Proximity to water also allows common leopards to conserve energy due to the concentration of animals and dense bushes/vegetation–making these locations favorable for ambushing prey species. A similar observation was reported by *Constant, Bell & Hill (2015)* in South Africa–most attacks on livestock occurred near water bodies (0–2 Km). Likewise, *Miller, Jhala & Jena (2016)* found livestock kills were most common between 2.9 ± 1 to 3.9 ± 0.2 km from water sources at the Kanha Tiger Reserve of Central India. Proximity to water is also related to the increased habitat suitability of common leopards (*Simcharoen et al., 2008*). Increased

probability of common leopard presence near water sources was also recorded in previous studies conducted in India (*Mondal, Sankar & Qureshi, 2013*) and Nepal (*Khaiju, 2017*). Similarly, *Abade (2013)* reported distance to water as the strongest predictor of livestock depredation risk (including common leopards) in Tanzania. Consistent with other studies, distance to water will likely increase the risk of common leopard depredation. However, *Naha et al. (2020)* found that livestock killings increased with distance from water bodies in the Himalayan region and argued that water availability might not be a limiting factor for HWC with carnivores in South Asia. Similarly, *Karanth et al. (2013)* found that distance to water is not strongly associated with livestock losses in the Western Ghats protected areas.

The risk of common leopard conflict peaks at the nearest distance from the road and human/livestock path. This aligns with the findings of *Sharma et al. (2020a)*, who reported a higher incidence of HWC within 5 kilometers of roads compared to farther distances. Another study by *Miller, Jhala & Jena (2016)* also discovered increased livestock kills within approximately 1.2 kilometers of roads in the Kanha Tiger Reserve, Central India. Contrary to our observations, common leopard attacks escalated beyond 8 kilometers from roads in South Africa (*Constant, Bell & Hill, 2015*). The reason for a higher probability of conflict near roads/paths in our study is likely due to rapid infrastructure development in the mid-hills of Nepal, including roads and walking trails (*Dhami et al., 2023c*), leading to habitat disruption and fragmentation, pushing common leopards closer to human settlements and grazing areas (*Dasgupta & Ghosh, 2015*; *Roy & Sukumar, 2017*).

Common leopards prefer living near the forest edge (*Lamichhane et al., 2019*; *Pokheral & Wegge, 2019*). Community forests in the Gorkha district are situated near villages (*Dhami et al., 2023c*). This inclination may increase negative interactions, especially with livestock depredation (*Kandel et al., 2023*). The study by *Lamichhane et al. (2018)* documented significant instances of livestock depredation occurring within a 500-meter radius of the boundaries of the forest and the village in CNP. In contrast, a higher frequency of livestock attacks close to forest edges was noted within CNP (*Ruda, Kolejka & Silwal, 2020*). Similarly, another report shows that the most common leopard attacks occurred within a 1-kilometer radius of forest edges in Baitadi District (*Baral et al., 2022a*). Furthermore, linear features such as roads and trails serve as linear corridors, influencing both the movement patterns of common leopards and the distribution of ungulate species, notably cattle, and goats, drawn by the availability of abundant forage in landscapes subjected to human exploitation. The increased potential of prey near roads amplifies the risk of common leopard attacks, given that people and animals utilize these well-defined paths frequently (*Neupane et al., 2022*; *Dhami et al., 2023d*).

The livestock depredation risk by common leopards was highest between 700 m and 800 m elevation. It can be related to more settlements in the study area and more livestock abundance at that elevation. Topographic factors (*i.e.,* terrain) frequently determine accessibility, which limits human-wildlife interactions (*Neilson, Nijman & Nekaris, 2013*). As mentioned by *Pitman, Swanepoel & Ramsay (2012)*, the probability of common leopard predation in South Africa increased by 6% at higher elevations where big cats often seek refuge. Maximum HWC was reported at 300–4000 m elevation at the Kanchenjunga landscape in Nepal (*Sharma et al., 2020a*) due to greater forest fragmentation at that

range. The risk of livestock kills was highest at lower elevations (670–780 m) and higher elevations (1540–1760 m) on Blouberg Mountain in South Africa (*Constant, Bell & Hill, 2015*). Likewise, the greater percentages of common leopard attacks on humans were reported at elevation ranges of 1,000 to 1,500 m and 100 to 500 m in Pauri Garhwal and North Bengal (India) (*Naha, Sathyakumar & Rawat, 2018*).

Based on our research findings and ensuing discussions, we suggest the expansion of this study to encompass other districts within Nepal where common leopards are prevalent. This extension is crucial for understanding depredation patterns in additional locations that our study did not address. While our research anticipated livestock depredation by considering various bio-physical factors, it did not delve into the identification of causative elements, including but not limited to habitat conditions, prey availability, herding and rearing practices, and the design of livestock sheds/corrals. The study sites are located in the goat farming pocket area where local farmers are involved in goat farming because of high demand and good price of small he-goats that are sacrificed by the devotees in *Gorkha Kalika and Manakamana* temples. Poorly constructed traditional corrals have been easily broken by the leopards during night time. Increasing emmigration trend of the viillages, fallow lands gradually changing in forested areas that contributed to support leopards' movements nearby human settlemnts and enough time to find the weaker goat corrals. Analysis of these factors is integral to formulating effective mitigation measures for fostering harmonious co-existence between humans and common leopards across the country. Further exploration and clarification in these areas are essential for informing and refining strategies to manage human-leopard conflict.

## CONCLUSIONS

This study highlighted issues and challenges for HLC management in Gorkha district, Nepal. Although our findings showed less area as high risk for livestock depredation by common leopards (which may be due to less coverage of the study area in the district), the entire district is susceptible due to the presence of community forests, increasing fallow lands and poor livestock rearing practices including poorly constructed corrals. The most important variables predicting risk were distance to water and roads, followed by elevation. Results suggest that stakeholders such as the District Forest Office and local government officials should develop HLC management strategies jointly and without delay. For example, DFO can faciliate to create a partnership scheme and generate subsidy fund from community forestry user groups, local government and provincial government to promote predator-proof corrals in the priority areas. Local government need to be responsible in local issues including HLC as they are clamming all natural resources are under their juridiction and utilization as well. There is an assumption that ''*jal* (water), *jamin* (land), *jungle* (forest), *jaributi* (non-timber forest products) are the property of *janata* (people). If so, it is not clear who the *janawar* (animals) belong to. Therefore, a budget should be allocated for HLC. Settlements near roads and water sources should become aware of these risks and focus on predator-proof corrals to reduce livestock losses. We suggest a holistic management plan for conflict management that includes many

stakeholders, such as local government administrative units, livestock owners/farmers, victims, academic scholars, and other experts. Furthermore, we recommend more studies to determine how fine-scale household characteristics can influence common leopard attacks on livestock and landscape features. This information can be used to formulate suitable management plans at the household level for the co-existence of villagers and common leopards that might be replicated in other regions of the country.

## ACKNOWLEDGEMENTS

Administrators at the Divisional Forest Office, Gorkha and Manaslu Conservation Area, provided livestock depredation records and logistical support. Prof. Mark Morgan, School of Natural Resources, University of Missouri, USA, provided valuable comments and edited the manuscript. Last but not least, we appreciate the information and cooperation provided by respondents during our household survey.

### Funding

We received research funds to complete our fieldwork from the Research Committee of Tribhuvan University, Institute of Forestry (IOF), Pokhara Campus, Nepal. The funders had no role in study design, data collection and analysis, decision to publish, or preparation of the manuscript.

### Grant Disclosures

The following grant information was disclosed by the authors:
Research Committee of Tribhuvan University, Institute of Forestry (IOF), Pokhara Campus, Nepal.

### Competing Interests

The authors declare there are no competing interests.

### Author Contributions

- Thakur Silwal conceived and designed the experiments, performed the experiments, analyzed the data, prepared figures and/or tables, authored or reviewed drafts of the article, co-ordination with concerned stakeholders during the research works, and approved the final draft.
- Bijaya Neupane conceived and designed the experiments, performed the experiments, analyzed the data, prepared figures and/or tables, authored or reviewed drafts of the article, field work coordination, and approved the final draft.
- Nirjala Raut conceived and designed the experiments, performed the experiments, prepared figures and/or tables, authored or reviewed drafts of the article, and approved the final draft.
- Bijaya Dhami performed the experiments, analyzed the data, prepared figures and/or tables, authored or reviewed drafts of the article, and approved the final draft.

- Binaya Adhikari performed the experiments, analyzed the data, prepared figures and/or tables, authored or reviewed drafts of the article, and approved the final draft.
- Amit Adhikari performed the experiments, analyzed the data, prepared figures and/or tables, authored or reviewed drafts of the article, and approved the final draft.
- Aakash Paudel performed the experiments, analyzed the data, prepared figures and/or tables, authored or reviewed drafts of the article, and approved the final draft.
- Shalik Ram Kandel performed the experiments, analyzed the data, prepared figures and/or tables, authored or reviewed drafts of the article, and approved the final draft.
- Mahamad Sayab Miya performed the experiments, analyzed the data, prepared figures and/or tables, and approved the final draft.

### Human Ethics

The following information was supplied relating to ethical approvals (*i.e.,* approving body and any reference numbers):

This study is only human participation for providing information rather than human tissue

### Data Availability

The raw data is available in the Supplemental File.

### Supplemental Information

Supplemental information for this article can be found online at http://dx.doi.org/10.7717/peerj.17497#supplemental-information.

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
