# Peer review of "Identifying risk zones and landscape features that affect common leopard depredation"

_PeerJ, doi:10.7717/peerj.17497_

## Round 0.1 · original submission · Major Revisions

Reviewers' comments on your work have now been received. The manuscript has been assessed by two reviewers. Reviewers indicated that the Introduction, the method, the result, and the discussion sections should be improved. Moreover, English editing and academic writing service is needed. I agree with this evaluation and I would, therefore, request for the manuscript to be revised accordingly.

Reviewer 1 has requested that you cite specific references. You may add them if you believe they are especially relevant. However, I do not expect you to include these citations, and if you do not include them, this will not influence my decision.

Reviewer 1 ·

Basic reporting

It was a pleasure to review the manuscript “Identifying risk zones and landscape features that affect leopard depredation in Nepal’s mid-hills” for PeerJ. I think this manuscript has great potential to be published in this journal but not in its current form. This paper looks at human-leopard conflict from a landscape ecology perspective by utilizing a common tool, MaxEnt software. However, justification of using this method was absent, as a lot of researcher use this method for its user-friendly interface and without having any modeling/ecological justification for it.
I left comments on some major flaws on the paper, and I would like to see it one more time before it gets published. I didn’t focus too much on minor comments as the manuscript will probably change significantly once major comments are applied. I think results of the paper and the graphs should be explained more in the text. The language of the manuscript should be revisited and corrected. Moreover, authors should increase the diversity of their citations as their paper has a great potential for generalizability to many situations all around the world, yet they mostly cited local papers. I provided a diverse list of citations that I suggest them using. After applying these comments, I should be in a better place to leave some minor and fine scale comments.

Experimental design

NA

Validity of the findings

NA

Additional comments

Major Comments:
For more clarity, the title can benefit from adding the word “livestock” before depredation to it.
Abstract:
-The abstract started talking write off the bat about human-leopard conflict. I’d suggest talking about conflict or at least human large carnivore conflict in the world, and then narrow it down to Nepal and Snow leopard problem.
-Please consider rewriting “Incidents on humans and livestock”.
- Please revise this claim “The Gorkha district is a critical but understudied location”. Use words that are less ambiguous and subjective instead of critical.
-You can use Panthera pardus as your keyword as well.
Introduction:

- Introduction is weak, authors start talking about leopards in Nepal which is quite specific. When you are submitting your paper to an international journal you need to take the global audience into account.
- Also the introduction lacks coherence and cohesion. Paragraphs and sentences should be linked to each other better.
- Also most of your citations are local, please cite more papers and from diverse countries and scientists. Also don’t limit your citations to leopards, I think you should talk about conflict as a general phenomena in the first paragraph, and then you can cite papers on carnivore conflict in general.
- Here are a few notable studies on leopards and their conflict from around the world that citing them will help your manuscript.
*Jacobson, A. P., Gerngross, P., Lemeris Jr, J. R., Schoonover, R. F., Anco, C., Breitenmoser-Würsten, C., ... & Dollar, L. (2016). Leopard (Panthera pardus) status, distribution, and the research efforts across its range. PeerJ, 4, e1974.
*Farhadinia, M. S., Ahmadi, M., Sharbafi, E., Khosravi, S., Alinezhad, H., & Macdonald, D. W. (2015). Leveraging trans-boundary conservation partnerships: Persistence of Persian leopard (Panthera pardus saxicolor) in the Iranian Caucasus. Biological Conservation, 191, 770-778.
*Kabir, M., Ghoddousi, A., Awan, M. S., & Awan, M. N. (2014). Assessment of human–leopard conflict in Machiara National Park, Azad Jammu and Kashmir, Pakistan. European Journal of Wildlife Research, 60, 291-296.
*Viollaz, J. S., Thompson, S. T., & Petrossian, G. A. (2021). When human–wildlife conflict turns deadly: comparing the situational factors that drive retaliatory leopard killings in south Africa. Animals, 11(11), 3281.
* Constant, N. L., Bell, S., & Hill, R. A. (2015). The impacts, characterisation and management of human–leopard conflict in a multi-use land system in South Africa. Biodiversity and Conservation, 24, 2967-2989.
*Babrgir, S., Farhadinia, M. S., & Moqanaki, E. M. (2017). Socio-economic consequences of cattle predation by the Endangered Persian leopard Panthera pardus saxicolor in a Caucasian conflict hotspot, northern Iran. Oryx, 51(1), 124-130.
*Khorozyan, I., Ghoddousi, A., Soofi, M., & Waltert, M. (2015). Big cats kill more livestock when wild prey reaches a minimum threshold. Biological Conservation, 192, 268-275.
* Athreya, V., Srivathsa, A., Puri, M., Karanth, K. K., Kumar, N. S., & Karanth, K. U. (2015). Spotted in the news: using media reports to examine leopard distribution, depredation, and management practices outside protected areas in Southern India. PLoS One, 10(11), e0142647.
Methods:
-How many people participate in your focus group? The focus group section is vague, please provide more information on how many people participated, how did you select them, and what was discussed.
-How 100m was selected as the threshold for incident uniqueness?
-You should talk about multicollinearity results, in the results section. Also in the methods section, you must mention it in chronological order. So, talk about the multicollinearity check and then talk about model fitness indices.
-It’s not clear to me what is distance to path. Please elaborate.
- Among variables, land use/land cover is not helpful at all, what types of land use/ land cover did you use?
-You mentioned none of your variables had a correlation of more than 0.7, but I can see in your table of variables that you used evi_sd/evi_max/evi_min/evi_mean. What was the justification of using them all? What do they mean for leopard ecology? And I suspect that they won’t have a correlation of less than 0.7.
-Why these variables were chosen in general also needs a better justification (leopard ecology-oriented).
- In your map of the study area, please show some neighboring countries too before zooming into your specific study location.

Results:
-Your result section is quite short and brief, please consider elaborating your results more and what they mean in the light of leopard ecology.
-Not sure if the colors you used for your risk zones map are color-blind friendly? I’d recommend using the blue-red or magma themes as the most common ones for showcasing the contrast in a -color-blind-friendly way.
-Please provide all response curves in the supplementary material as well.
-You need to report the trend of your response curves and then discuss why that’s the case.
For example, your response curve is telling you that leopard conflict decreases with increasing distance from water, but after your interpretation I thought the other direction was observed (the closer to water bodies, the higher the conflict).
Discussion:
Your discussion section can benefit a lot from more coherent and cohesive writing. In many paragraphs, you basically reported your results again followed by someone else findings, without synthesizing at the end. You need to make sense of your results in the light of other works around the world. Your results are either in line with other studies or are contradictory to other people’s results. In both scenarios, you need to talk about synthesis and conclude sth from it.

Minor Comments:
-Please avoid using abbreviations in the abstract.
- Line 26: When you are referring to MaxEnt for the first time, provide the full name for it (Maximum Entropy).
-Abstract: Results part is quite interesting but recommendations can be improved and tailored more to your findings. I’ll read on.
-Line 88: You can cite this article that used MaxEnt as well:
Nayeri, D., Mohammadi, A., Hysen, L., Hipólito, D., Huber, D., & Wan, H. Y. (2022). Identifying human-caused mortality hotspots to inform human-wildlife conflict mitigation. Global Ecology and Conservation, 38, e02241.
-Line 91-93: This belongs to the Methods section.
-Line 115-116: “The predominant livelihood strategy is agriculture, especially farming and livestock husbandry”. Is livestock husbandry considered agriculture? I don’t think so.

-Line 118-121: Please provide the English name for floral species as well. Also, avoid using etc in scientific writing.

-Line 121: Please change “Major wild faunal species”, the main wildlife species in the area are…

-Line 347: Please correct “academicians” to academics or scholars.

·

Basic reporting

Please see detailed comments below

Experimental design

Please see detailed comments below

Validity of the findings

Please see detailed comments below

Additional comments

Thank you for giving me the chance to review this interesting paper that aims to identify risk zones and landscape features that affect leopard depredation in the mid hills of Nepal. I think the data and approach used by the author is useful in delimiting these important “conflict” regions. While I think this is a useful paper, I do think there are several nuances needed before this can be accepted. I highlight them below and hope you find these useful:

Abstract: Be careful between human-wildlife conflict and human-wildlife interactions that are negative (eg. Redpath et al., 2015: Redpath, S. M., Bhatia, S., & Young, J. (2015). Tilting at wildlife: reconsidering human–wildlife conflict. Oryx, 49(2), 222-225.). It would be nice to have this nuance throughout the paper.

Introduction: I would recommend that rather than starting with the leopard and then moving to the theme of HWC, especially in Nepal, do it the other way around. It would be nice to briefly speak about human-wildlife interactions (not conflict) and then talk a bit about how there are negative interactions which are terms as HWC and then come to the specific case of the leopards in Nepal’s mid-hills. See also Bhatia et al. 2020: Bhatia, S., Redpath, S. M., Suryawanshi, K., & Mishra, C. (2020). Beyond conflict: exploring the spectrum of human–wildlife interactions and their underlying mechanisms. Oryx, 54(5), 621-628.) which talks about HWC as a spectrum rather than either/or. Again getting this nuance into both your introduction/discussion and also results is important

Line 56: “As a habitat generalist…”

Material and Methods:
Line 101-105: Just wondering why the study was limited to such a small percentage of the Gandaki province?

Line 131-133: It would nice to elaborate a bit on the what provisions there are by different government agencies to “compensate/provide relief” for victims?

Line 143: how many FDGs were done? What was the composition of each? See Nyumba et al. (2018) (O. Nyumba, T., Wilson, K., Derrick, C. J., & Mukherjee, N. (2018). The use of focus group discussion methodology: Insights from two decades of application in conservation. Methods in Ecology and evolution, 9(1), 20-32.) on the key insights for use of FDGs. Were these followed? If not, do discuss any limitation in the discussion section

Line 158: Was the time of the event also recorded? (Season and day time) as I feel that is also a factor impacting depredation. If not this needs explicit discussion later.
Table 1: In table it will be very useful to add another column describing the predicted relationship of each variable with your model output. This will help readers see the reasoning in using those variables. Additionally, and importantly, did you check for auto-correlation between each variable? This would have been useful to ensure you are not using variables that might ecologically mean similar things.

Line 211-212: I see you have looked at multicollinearity which is great! Can you add the process and results in supplementary material, please?
Results: I wonder if the authors are missing out on some nuance by only conducting the MaxEnt analysis. Was there some interesting “micro-scale” information that was received through the FDGs about leopard depredations? It would be great if you could add, even if qualitatively, what was said in the FDGs.

Results: Also, is it not possible to look at leopard attacks based on livestock type? I would imagine this would change the results and the predictors? There might be socio-economic reasons for people to keep certain type of livestock (eg. Small-bodied) over others (eg. Large bodied) and that might impact depredation process.

Model accuracy: can you justify using reference, why the quantities you are quoting are “good”

Discussion – risk zones with respect to leopard attacks – is there some data on prey apart from overall trends? Also, what about dogs and their importance as prey for leopards? In many parts of India, dogs are the main prey for leopards and they (dogs) might then impact how leopards interact with humans/livestock. See for example Athreya et al., 2016 (Athreya, V., Odden, M., Linnell, J. D., Krishnaswamy, J., & Karanth, K. U. (2016). A cat among the dogs: leopard Panthera pardus diet in a human-dominated landscape in western Maharashtra, India. Oryx, 50(1), 156-162.)

Discussion – Important environmental variables predicting the leopard risk zone: While you have done a great job of referencing different studies and discuss similar results from other systems, I think it would be valuable to have a short discussion about how socio-economic changes in your system (example more development leading to, perhaps, more roads being made), maybe impacting human-wildlife conflict with leopards.

Discussion: I would like to see some more discussion on the caveats of your studies, especially your modelling approach.

Conclusion: Line 342-344: I think you are missing a point here: Local stakeholders are also the herders and villagers, whereas the DFOs and local government officials, are indeed local but more administrative.

- Munib Khanyari

---

## Round 0.2 · Minor Revisions

Highlight the remaining issues within the manuscript.

Reviewer 1 ·

Basic reporting

I reviewed the response to reviewers as well as the revised manuscript.

Kudos to authors for all the effort and time in substantially editing their draft, it looks much better now.

There are still a few items to be taken care of:

- While the new citations are great, they should be placed at the end of the sentences. Right now, many of the citations are at the middle of a sentence and impede the flow and make readers job hard!
- Please avoid using "etc" in scientific writing (Line 137)
- There are still some language errors across the manuscript that need to be taken care of (e.g. biasness)
-Please avoid starting the sentence with an abbreviation (e.g. EVI)
- I still think their management recommendations are too general and can be said about any other study!
- Paragraphs in the discussion especially are heterogenous in their length and can be better linked!

After these being taken care of, I think the manuscript can be accepted!

Experimental design

NA

Validity of the findings

NA

Additional comments

NA

---

## Round 0.3 · accepted · Accept

Please make a final check of the manuscript as requested by Reviewer 1.

Reviewer 1 ·

Basic reporting

I think the authors did a satisfactory job in addressing the comments raised in the second round of review. The article is ready to be published. However still a few English issues with the new paragraphs they added which I'm sure it will be addressed in the proof.

Experimental design

NA

Validity of the findings

NA

Additional comments

NA